



# Estimation of isentropic stirring and mixing and their diagnosis for the stratospheric polar vortex

Zhiting Wang[1], Nils Hase[2], Wenshou Tian[1], Mengchu Tao[3]

1. College of Atmospheric Science, Lanzhou University, 730000 Lanzhou, China

2. Institute of Environmental Physics, University of Bremen, Bremen, Germany

3. Carbon neutrality research center, Institute of Atmospheric Physics, Chinese Academy of Science, China

*Correponding author:* Zhiting Wang, wangzt@lzu.edu.cn





## Abstract

Isentropic stirring and mixing are important processes that determine the distribution of long-lived trace gases in the stratosphere. Stirring stretches tracer contours into filaments and mixing dissipates tracer variance. The combined effects on tracer transport by stirring and mixing are quantified by the effective diffusivity in the modified Lagrangian-mean (MLM) theory that diagnoses tracer transport in an areal coordinate. Here a method is developed to diagnose transport processes based

on tracer contours in geographic coordinates. Compared to the MLM theory this method has resolving ability along tracer contours and quantifies stirring and mixing separately. Also, the influence of diabatic motion on the diagnosed stirring is reduced, which is useful for stratospheric analysis where diabatic motion is uncertain. The developed method is validated in a methane simulation experiment. The diagnosed stirring effects are consistent with established knowledge

about stratospheric dynamics. Finally, stirring and mixing effects on trace gases in the polar vortex are diagnosed during a northern polar vortex period. According to the diagnosis stirring and mixing always increase the methane concentration in the polar vortex. However, their effects are reversed by vortex movement and deformation in most cases. Only in a few cases, planetary waves can penetrate into the vortex and stirring increases the methane concentration in the vortex. The

developed method is readily applicable to diagnose stratospheric transport processes from satellite observed trace gas distributions.

## 1 Introduction

Distributions of long-lived trace gases in the stratosphere are determined mostly by the residual circulation and isentropic stirring and mixing (Haynes, 2004). Air following the residual circulation

ascends around the tropics, moves polewards and descends in high latitudes. Most long-lived trace gases have chemical sinks in the stratosphere. These chemical sinks decay the tracer and produce gradients in its distribution that reflect the residence time of the air mass in the stratosphere. As a result, the residual circulation always tends to produce a tropics-to-pole gradient in the tracer distribution on each isentropic surface. Along isentropic surfaces, wave disturbances, e.g. planetary

waves, stretch and deform the contours of the trace gas. This process is referred to as isentropic stirring. Mixing describes the effect of small-scale motions such as turbulent and molecular





diffusion that lead to an irreversible fusion of air masses. Stirring can enhance mixing by producing long, filamentary contours and stronger gradients across the contour. In contrast to the residual circulation, isentropic stirring and mixing tend to decline the tropics-to-pole gradient in the trace

gas distributions.

One of the theories that describe these physical processes is the modified Lagrangian-mean (MLM) framework (Nakamura, 1995 and 1996). The MLM approach defines an areal/mass coordinate for each value of the mixing ratio as the area/mass enclosed by the corresponding contour. In this way, a functional relationship between the mixing ratio and the areal coordinate can be established. In the

two-dimensional case, e.g. along isentropic surfaces, the advection-diffusion equation with respect to spatial coordinates transforms into a one-dimensional diffusion equation with respect to the areal coordinate. In this diffusion equation the combined effect of stirring and mixing is represented by a single parameter referred to as effective diffusivity. It consists of the constant diffusivity of the advection-diffusion equation and a variable scaling referred to as equivalent length. The diffusivity

can be estimated by means of the tracer variance equation (Allen and Nakamura, 2001).

The effective diffusivity has been interpreted to diagnose transport barriers and regions with strong mixing in the atmosphere and the ocean (Nakamura and Ma, 1997; Haynes and Shuckburgh, 2000; Allen and Nakamura, 2001; Marshall et al. 2006; Abernathy et al. 2010). One advantage of the MLM theory is that fluxes along the areal coordinate are diagnosed based on the contour of the

tracer without using the wind field. This characteristic is valuable in stratospheric applications since accurate wind information are much sparser than trace gas observations in the stratosphere. Further developments of the MLM theory include partitioning of the fluxes into opposing directions (Nakamura, 2004), incorporating diabatic diffusion into the effective diffusivity (Leibensperger and Plumb, 2014) and extending the concept of effective diffusivity to the troposphere (Chen and

Plumb, 2014).

A shortcoming of the MLM theory is that the diagnosis based on the areal/mass coordinate does not have resolvability along the tracer contours. This aspect is inconvenient when the theory is applied to trace gases in the stratosphere whose mixing ratios usually present extremes in the tropics. A separate application of the MLM theory to each hemisphere leads to confusion in the tropics since

the rising branch of the residual circulation moves North and South during a year. Moreover, trace gases like water vapor and ozone have additional minima in the polar vortex, especially the southern one. In this case, diagnosis defined on the contour of trace gases are averages over well separated regions and not easy to interpret. Attempts to adapt the Lagrangian-mean formalism to a regional mixing diagnostic include Nakamura (2001) and D'Ovidio et al. (2009). The diagnostic



'mixing efficiency' proposed in Nakamura (2001) is the Eulerian-mean counterpart to the equivalent length in the MLM theory. The mixing efficiency relies on a spatial averaging operator over latitudes and longitudes and only represents stirring processes with scales smaller than those of the averaging operator. In contrast, the equivalent length includes all scales above turbulence. The diagnostic by D'Ovidio (2009) combines the tracer-based effective diffusivity and the particle-

based Lyapunov exponent calculation and requires more meteorological data than the MLM method.

In this study, we develop a method that diagnoses transport processes with respect to the geographic coordinate based on the tracer contour. The diagnosis has resolvability along the tracer contour and differentiates stirring and mixing. Stirring and mixing have different roles in distributing trace gases

in the stratosphere and are controlled by horizontal wind and turbulent processes, respectively. Also, the developed method reduces the influence of diabatic motion on the diagnosed stirring, which is advantageous for stratospheric analysis where observations of diabatic heating are rare.

The remainder of the article is organized as follows: the method is developed and described in Sect. 2. In Sect. 3 we describe the numerical setup that we use as a test case for our method. The analysis

of this numerical scenario is described in Sect. 4. A discussion of the method follows in Sect. 5 before we close with a conclusion.

## 2 Method

In the stratosphere trace gas contours along isentropic surfaces are modified continuously by horizontal motion along the isentropic surfaces, diabatic motion across the isentropic surfaces,

diffusion and chemical sources and sinks. In this section we derive an expression that describes the stretching and deformation of the contour by horizontal motion along isentropic surfaces. We refer to this quantity as (isentropic) stirring. The quantity is based on the evaluation of the contour and does not require isentropic wind information. The derivation of the diagnostic formulas for isentropic stirring is based on the modified Lagrangian-mean framework (e.g. Nakamura, 1995),

which we briefly review in the following paragraph.

### 2.1 Modified Lagrangian-mean framework

Consider a long-lived atmospheric trace gas distribution with mixing ratio $q = q(x, y, \theta, t)$, where $x$ and $y$ are geographic coordinates, $\theta$ is the vertical potential temperature coordinate and $t$ is time. The temporal evolution of $q$ is described by the advection-diffusion equation





$$\frac{\partial q}{\partial t}=-\vec{v}\cdot\nabla q-\dot{\theta}\frac{\partial q}{\partial \theta}+\dot{q} \quad , \quad (1)$$

where the first term describes advection by isentropic wind along the isentropic surface, the second term describes advection by diabatic (vertical) motion $\dot{\theta}$ and $\dot{q}$ summarizes all nonconservative processes, particularly turbulent and molecular diffusive mixing, sources and sinks.

Following the concept of the modified Lagrangian-mean (Nakamura, 1995) the density-weighted areal integral of a function $f$ over the the area enclosed by the contour line $q$ on the isentropic surface $\theta$ is defined by

$$M(f)(q,\theta,t)=\iint\limits_{q'(x,y,\theta,t)\leq q} f(x,y,\theta,t)\,\sigma d(x,y) \quad ,$$

where $\sigma=-g^{-1}\partial p/\partial \theta$ is the potential temperature coordinate density and $d(x,y)$ is an
infinitesimal area on the *xy*-plane. Note that the mass enclosed by the contour line and the isentropic surfaces $\theta$ and $\theta+\delta\theta$ is just $m\delta\theta$ , where *m* is

$$m(q,\theta,t)=M(1)(q,\theta,t) \quad .$$

Generally, the relationship between *m* and *q* on an isentropic surface $\theta$ at any time *t* a is one-to-one mapping. This property allows to establish the inverse relationship, i.e. $q=q(m,\theta,t)$ . As a
consequence, functions of *q* can also be interpreted as functions of *m*, e.g.

$$M(f)(m,\theta,t)=M(f)(q(m,\theta,t),\theta,t) \quad .$$

For our derivation the function $m(x,y,\theta,t)=m(q(x,y,\theta,t),\theta,t)$ will be of special importance. It can be interpreted as scaled version of *q* with the time-varying scaling given by the relation $m(q,\theta,t)$ (see Fig. 1 for an illustration).

The modified Lagrangian-mean approach observes the evolution of the atmospheric tracer with respect to this mass coordinate *m*, rather than in geographic coordinates *x* and *y*. The evolution equation of *q* with respect to *m* is given by (Nakamura, 1995, Eq. 2.7 (a,b,c))

$$\frac{\partial q}{\partial t}\big|_{m,\theta}=\left(\frac{\partial M(\dot{q})}{\partial q}\big|_{\theta,t}+\frac{\partial M(\dot{\theta})}{\partial \theta}\big|_{m,t}\right)\frac{\partial q}{\partial m}\big|_{\theta,t}-\hat{\theta}\frac{\partial q}{\partial \theta}\big|_{m,t} \quad , \quad (2)$$

where $\hat{\theta}=\frac{\partial M(\dot{\theta})}{\partial m}\big|_{\theta},t$ is a weighted average with weighting corresponding to the mass between
the contour and a neighbouring contour with an infinitesimally different value of *m*. It states that the relation *q(m)* in an isentropic layer changes by nonconservative processes $\dot{q}$ and diabatic motion





$\dot{\theta}$ . For a volume bounded by the contour and two isentropic surfaces, the first term on the right hand side represents the diffusive flux across the bounding contour while the other two describe the divergence of diabatic mass flux and vertical transport.

In the following, $q$ is referred to different sets of variables, i.e. $(x,y,\theta,t)$ , $(q,\theta,t)$ or $(m,\theta,t)$ whenever required. The overline operator, $\overline{(f)}=\int f\,d(x,y)/\int d(x,y)$ , denotes the global area-weighted average over an isentropic surface.

### 2.2 Derivation of the diagnostic method

Based on the Lagrangian-mean framework, we derive expressions for isentropic stirring and

mixing. After a partly technical derivation we present the final forms of both quantities at the end of this section.

The derivation starts by expressing the Eulerian partial time-derivative in Lagrangian-mean coordinates, i.e.

$$\frac{\partial q(x,y,\theta,t)}{\partial t}\big|_{x,y,\theta}=\frac{\partial q(m,\theta,t)}{\partial m}\big|_{\theta,t}\frac{\partial m}{\partial t}\big|_{x,y,\theta}+\frac{\partial q(m,\theta,t)}{\partial t}\big|_{m(x,y,\theta,t),\theta} \quad . \quad (3)$$

The left hand side of the equation denotes local changes to the trace gas mixing ratio $q$ described by the processes of the Eulerian evolution equation (1). The second term on the right hand side only represents processes that change the distribution of $q$ with respect to the mass coordinate in the Lagrangian-mean evolution equation (2). These nonconservative processes include sources and sinks, diffusive mixing and advection by air flow across isentropic surfaces. Among other

processes, isentropic stirring is included in the term $\frac{\partial m}{\partial t}\big|_{x,y,\theta}$ . To derive an expression for stirring, we rearrange the equation and replace the Eulerian and modified Lagrangian-mean time derivatives with the right hand sides of the evolution equation (1) and (2), respectively. We have

$$\frac{\partial m}{\partial t}\big|_{x,y,\theta}=(\frac{\partial q}{\partial t}\big|_{x,y,\theta}-\frac{\partial q}{\partial t}\big|_{m,\theta})\frac{\partial m}{\partial q}\big|_{\theta,t}$$
$$=[-\vec{v}\cdot\nabla q-\dot{\theta}\frac{\partial q}{\partial\theta}\big|_{x,y,t}+\dot{q}-(\frac{\partial M(\dot{q})}{\partial q}\big|_{m,t}+\frac{\partial M(\dot{\theta})}{\partial\theta}\big|_{m,t})\frac{\partial q}{\partial m}\big|_{\theta,t}+\hat{\theta}\frac{\partial q}{\partial\theta}\big|_{m,t}]\frac{\partial m}{\partial q}\big|_{\theta,t} \quad , \quad (4)$$

where $q$ is a function of either $(x,y,\theta,t)$ or $(m,\theta,t)$ wherever needed. As described, $\vec{v}$

denotes the isentropic wind, $\nabla$ the gradient operator within isentropic surfaces, $\dot{\theta}$ the diabatic motion and $\dot{q}$ nonconservative processes such as diffusive mixing, sources and sinks. We further manipulate the equation using





$$\frac{\partial M(\dot{q})}{\partial q}\big|_{\theta,t}\frac{\partial q}{\partial m}\big|_{\theta,t}=\frac{\partial M(\dot{q})}{\partial m}\big|_{\theta,t}\equiv\hat{\dot{q}}\quad,\quad\frac{\partial m}{\partial q}\big|_{\theta,t}(\vec{v}\cdot\nabla q)=\vec{v}\cdot\nabla m\quad\text{and}$$

$$\frac{\partial q(x,y,\theta,t)}{\partial\theta}\big|_{x,y,t}=\frac{\partial q}{\partial m}\big|_{\theta,t}\frac{\partial m(x,y,\theta,t)}{\partial\theta}\big|_{x,y,t}+\frac{\partial q}{\partial\theta}\big|_{m(x,y,\theta,t),t}\quad.$$

With these changes Eq. 4 transforms to

$$\frac{\partial m}{\partial t}\big|_{x,y,\theta}=-\vec{v}\cdot\nabla m-\dot{\theta}\frac{\partial m}{\partial\theta}\big|_{x,y,t}-\frac{\partial M(\dot{\theta})}{\partial\theta}\big|_{m,t}-(\dot{\theta}-\hat{\dot{\theta}})\frac{\partial q}{\partial\theta}\big|_{m,t}\frac{\partial m}{\partial q}\big|_{\theta,t}+(\dot{q}-\hat{\dot{q}})\frac{\partial m}{\partial q}\big|_{\theta,t}\quad,\text{(5)}$$

The equation describes the processes that modify the contour of *m*. These are motion along isentropic surfaces, horizontal divergence motion due to air expansion and convergence of diabatic mass flux, diabatic advection, and diffusion and chemical reactions, respectively.

The goal is to isolate processes that describe isentropic stirring from those that describe nonconservative and diabatic processes. Recall that we refer to stirring as contour deformation due to motion along isentropic surfaces. The effects by nonconservative processes and diabatic advection should be subtracted from both sides of the equation. After this manipulation there is

$$\frac{\partial m}{\partial t}\big|_{x,y,\theta}-\frac{\partial m}{\partial q}\big|_{\theta,t}(\dot{q}-\hat{\dot{q}})+(\dot{\theta}-\hat{\dot{\theta}})\frac{\partial q}{\partial\theta}\big|_{m,t}\frac{\partial m}{\partial q}\big|_{\theta,t}$$
$$=-\vec{v}\cdot\nabla m-\dot{\theta}\frac{\partial m}{\partial\theta}\big|_{x,y,t}-\frac{\partial M(\dot{\theta})}{\partial\theta}\big|_{m,t}\quad.\text{(6)}$$

It is clear that modification to $m(x,y,\theta,t)$ by nonconservative processes and diabatic advection occurs only when these processes are nonuniform along the contour of *q*. Eq. 6 is an expression for the contour modification caused by processes other than diffusion, chemical reactions and diabatic advection. Except for vertical advection, diabatic motion can also modify $m(x,y,\theta,t)$ by changing mean air density over isentropic surfaces. The mean air density variation does not occur

explicitly on the right-hand side of Eq. 6. In the following, further analysis is conducted to show that the right-hand side of Eq. 6 consists of horizontal advection along isentropic surfaces and effects by mean air density variations only.

By definition of the operator *M* and by use of the Leibniz integral rule we have

$$\frac{\partial M(\dot{\theta})}{\partial\theta}\big|_{m,t}=\frac{\partial}{\partial\theta}\iint\limits_{q'(x,y,\theta,t)\leq q(m,\theta,t)}\sigma\dot{\theta}d(x,y)\big|_{m,t}$$
$$=\iint\limits_{q'(x,y,\theta,t)\leq q(m,\theta,t)}\frac{\partial\sigma\dot{\theta}}{\partial\theta}\big|_{x,y,t}d(x,y)-\oint\limits_{q'(x,y,\theta,t)=q(m,\theta,t)}\sigma\dot{\theta}\frac{\partial m}{\partial\theta}\big|_{x,y,t}\frac{1}{|\nabla m|}dl\quad.\text{(7)}$$





In Eq. 7  $\frac{\partial \sigma \dot{\theta}}{\partial \theta}|_{x,y,t}$  contains contributions by horizontal divergent motion caused by vertical convergence of diabatic motion and expansion of air, e.g. due to decreasing air density of uplifting air. Density changes are described by the mass continuity equation. Expressed in the potential temperature coordinate it reads

$$\frac{\partial \sigma}{\partial t}|_{x,y,\theta} + \nabla \cdot \sigma \vec{v} + \frac{\partial \sigma \dot{\theta}}{\partial \theta}|_{x,y,t} = 0 \quad .$$

Applying the areal average operator over an isentropic surface, which we denote by an overline, gives

$$\frac{\partial \overline{\sigma}}{\partial t}|_{\theta} + \frac{\partial \overline{\sigma \dot{\theta}}}{\partial \theta}|_{t} = 0 \quad .$$

This relation indicates that change in the mean density of an isentopic surface is related to diabatic mass divergence. The horizontal divergent wind  $\vec{v}_d$  that compensates the local diabatic mass

convergence and the expansion of air in an isentropic layer is characterized by

$$\nabla \cdot \sigma \vec{v}_d + \frac{\partial \sigma \dot{\theta}}{\partial \theta}|_{x,y,t} - \frac{\partial \overline{\sigma \dot{\theta}}}{\partial \theta}|_{t} = 0 \quad . \quad (8)$$

In combination with the condition  $\nabla \times \sigma \vec{v}_d = 0$  the horizontal divergent wind is uniquely defined. We use the horizontal divergent wind to reinterpret some terms in Eq. 6. A manipulation of Eq. 6 reads

$$\frac{\partial m}{\partial t}|_{x,y,\theta} - \frac{\partial m}{\partial q}|_{\theta,t}(\dot{q} - \frac{\partial M(\dot{q})}{\partial m}|_{\theta,t}) + (\dot{\theta} - \hat{\dot{\theta}})\frac{\partial q}{\partial \theta}|_{m,t}\frac{\partial m}{\partial q}|_{\theta,t}$$
$$= -(\vec{v} - \vec{v}_d) \cdot \nabla m \underbrace{- \vec{v}_d \cdot \nabla m - \dot{\theta}\frac{\partial m}{\partial \theta}|_{x,y,t}}_{D} - \frac{\partial M(\dot{\theta})}{\partial \theta}|_{m,t} \quad . \quad (9)$$

The term defined as D describes advection of  $m(x,y,\theta,t)$  by the horizontal divergent wind reduced by the horizontal expansion associated with vertical diabatic motion. As a result, the term represents the effect of convergence of diabatic motion. For further interpretation note that



$$\frac{\partial M(D)}{\partial m}\Big|_{\theta,t}=\frac{\partial}{\partial m}\iint_{q'(x,y,\theta,t)\leq q(m,\theta,t)}\sigma(-\vec{v}_d\cdot\nabla m-\dot{\theta}\frac{\partial m}{\partial\theta}\big|_{x,y,t})d(x,y)\big|_{\theta,t}$$

$$=\frac{1}{\delta m}\oint_{q'(x,y,\theta,t)=q(m,\theta,t)}\sigma(-\vec{v}_d\cdot\nabla m-\dot{\theta}\frac{\partial m}{\partial\theta}\big|_{x,y,t})\frac{\delta m}{|\nabla m|}dl$$

$$=\oint_{q'(x,y,\theta,t)=q(m,\theta,t)}-\sigma\vec{v}_d\cdot\vec{n}_m dl-\oint_{q'(x,y,\theta,t)=q(m,\theta,t)}\sigma\dot{\theta}\frac{\partial m}{\partial\theta}\big|_{x,y,t}\frac{1}{|\nabla m|}dl$$

$$=\iint_{q'(x,y,\theta,t)\leq q(m,\theta,t)}(-\nabla\cdot\sigma\vec{v}_d)d(x,y)-\oint_{q'(x,y,\theta,t)=q(m,\theta,t)}\sigma\dot{\theta}\frac{\partial m}{\partial\theta}\big|_{x,y,t}\frac{1}{|\nabla m|}dl$$

$$=\iint_{q'(x,y,\theta,t)\leq q(m,\theta,t)}\frac{\partial\overline{\sigma}}{\partial t}\big|_\theta d(x,y)+\iint_{q'(x,y,\theta,t)\leq q(m,\theta,t)}\frac{\partial\sigma\dot{\theta}}{\partial\theta}\big|_{x,y,t}d(x,y)-\oint_{q'(x,y,\theta,t)=q(m,\theta,t)}\sigma\dot{\theta}\frac{\partial m}{\partial\theta}\big|_{x,y,t}\frac{1}{|\nabla m|}dl$$

$$=S_m\frac{\partial\overline{\sigma}}{\partial t}\big|_\theta+\frac{\partial M(\dot{\theta})}{\partial\theta}\big|_{m,t}$$

where $\vec{n}_m$ is the unit vector parallel to the horizontal gradient of $m(x,y,\theta,t)$ within isentropic surfaces and $\delta m$ is a infinitesimal perturbation of $m$. $S_m$ denotes the area enclosed by the contour line $q(m)$. We use the above relation and subtract the term $S_m\frac{\partial\overline{\sigma}}{\partial t}\big|_\theta$ from both sides of Eq. 9 to receive an expression for isentropic stirring

$$\frac{\partial m}{\partial t}\Big|_{x,y,\theta}^{stir}=\frac{\partial m}{\partial t}\big|_{x,y,\theta}-\frac{\partial m}{\partial q}\big|_{\theta,t}(\dot{q}-\hat{q})+(\dot{\theta}-\hat{\theta})\frac{\partial q}{\partial\theta}\big|_{m,t}\frac{\partial m}{\partial q}\big|_{\theta,t}-S_m\frac{\partial\overline{\sigma}}{\partial t}\big|_\theta \qquad (10a)$$

$$=-(\vec{v}-\vec{v}_d)\cdot\nabla m+D-\hat{D} \qquad (10b)$$

The first line represents the diagnostic formula for the stirring effect and the second line explains its physical content.

The first term, $-(\vec{v}-\vec{v}_d)\cdot\nabla m$ , describes the advection of $m(x,y,\theta,t)$ by horizontal wind other than the horizontal divergent wind induced by convergence of diabatic motion. Particularly, the wind $\vec{v}-\vec{v}_d$ includes zonal mean zonal wind and wave-related wind. The zonal mean zonal

wind can stretch spikes of the contour produced by wave disturbances. The term $D-\hat{D}$ describes the nonuniform part of the horizontal motion $D$ associated with vertical convergent motion along the contour. The motion included in both terms modifies $m(x,y,\theta,t)$ but conserves the mass enclosed by the contour $q$ and is identified as stirring.

By including the definition of isentropic stirring (Eq. 10) and the evolution equation with respect to

$m$ (Eq. 2), Eq. 3 transforms to

$$\frac{\partial q}{\partial t}\Big|_{x,y,\theta}=\frac{\partial q}{\partial m}\big|_{\theta,t}(\frac{\partial m}{\partial t}\big|_{x,y,\theta}^{stir}+\frac{\partial m}{\partial t}\big|_{x,y,\theta}-\frac{\partial m}{\partial t}\big|_{x,y,\theta}^{stir})+\frac{\partial q}{\partial t}\big|_{m,\theta}$$

$$=\frac{\partial q}{\partial m}\big|_{\theta,t}\frac{\partial m}{\partial t}\big|_{x,y,\theta}^{stir}+\dot{q}-\dot{\theta}\frac{\partial q}{\partial\theta}\big|_{m,t}+(\frac{\partial M(\dot{\theta})}{\partial\theta}\big|_{m,t}+S_m\frac{\partial\overline{\sigma}}{\partial t})\frac{\partial q}{\partial m}\big|_{\theta,t} \qquad . \quad (11)$$





The right hand side terms in the second line describe the effects of stirring, nonconservative processes, temporal variation of $q$ caused by vertical motion and mean expansion of air, respectively. For a long-lived trace gas with negligible stratospheric sources and sinks, e.g.

methane, molecular and turbulent diffusive mixing are the only nonconservative processes. We neglect the effect of diabatic diffusion and set $\dot{q} = k_h \nabla_h^2 q$ with the isentropic diffusivity $k_h$. The effects of isentropic stirring and mixing on the trace gas concentration $q$ are

$$\frac{\partial q}{\partial t}\Big|_{x,y,\theta}^{stir} = \frac{\partial q}{\partial m}\Big|_{\theta,t} \frac{\partial m}{\partial t}\Big|_{x,y,\theta}^{stir}, \quad \frac{\partial q}{\partial t}\Big|_{x,y,\theta}^{mix} = k_h \nabla^2 q \quad . \quad (12)$$

Both diagnostic quantities, isentropic stirring and mixing, are based on the evaluation of the trace
gas field $q$ and its scaled version m and do not require isentropic wind information. A procedure to estimate the isentropic diffusivity $k_h$ is presented in Section 2.4.

## 2.3 Remarks to isentropic stirring and mixing

The stirring effect, $\frac{\partial q}{\partial t}\Big|_{x,y,\theta}^{stir}$, is diagnosed as residual between local temporal variation of $m$,

$\frac{\partial m}{\partial t}\Big|_{x,y,\theta}$, and the nonconservative effects scaled by the function $q(m)$. The same residual can be

directly calculated through the evolution equation of $q$ in Eulerian coordinates (Eq. 1) instead of that of $m$ (Eq. 5). However, the local temporal variation of $m$ includes less contribution of diabatic motion compared to that of $q$. Only the nonuniform part of diabatic heating along the contour, $\dot{\theta} - \hat{\dot{\theta}}$, can lead to variation of $m$. The reduced influence of diabatic data is useful for stratospheric applications where observations of diabatic heating are sparse. Some examples that

show such a reduction are given in Sect. 4. In addition, $\frac{\partial m}{\partial t}\Big|_{x,y,\theta}$ does not include the

contribution of uniform divergent motion across the contour, unlike $\frac{\partial q}{\partial t}\Big|_{x,y,\theta}$. Examples of such uniform divergent motion are the horizontal branches of the residual circulation. The uniform divergent motion can not be removed in an analysis based on the residual of local temporal variation of $q$ and its nonconservative effects. From a practical viewpoint, the influence of

observation errors on $m$ is smaller than on $q$ since $m$ depends on the relative distribution of tracer mixing ratios only.

Similarly, only nonuniform diffusion along the contour can modify $m$. In cases where tracer isolines almost parallel latitudinal circles, e.g. in the tropics, the summer hemisphere and nearby the





boundary of a stable polar vortex, diffusive effects on $m$ are reduced compared to those on $q$. The

influence of an inaccurately estimated diffusivity is then reduced.

### 2.4 Estimation of isentropic diffusivity

The isentropic and diabatic diffusivities $k_h$ and $k_\theta$ describe the diffusive mixing by molecular and turbulent diffusion along and vertical to the isentropic surfaces. Turbulent diffusion parameterizes the mixing effect from unresolved small scale processes. Global estimates can be inferred from the

tracer variance equation (e.g. Leibensperger and Plumb, 2014). Wang et al (2020) extended the method to estimate the average diffusivities for a stratospheric region bounded by two isentropic surfaces $\theta_1$ and $\theta_2$ by means of the spatially bounded tracer variance equation

$$
\begin{aligned}
&\frac{1}{2}\frac{d\overline{q^2}}{dt} - \frac{1}{2}\iint_{\theta=\theta_1}\sigma\dot\theta q^2\frac{d(x,y)}{M} + \frac{1}{2}\iint_{\theta=\theta_2}\sigma\dot\theta q^2\frac{d(x,y)}{M} \\
&= -k_\theta\left(\frac{1}{2}\iint_{\theta=\theta_1}\sigma\frac{\partial q^2}{\partial\theta}\frac{d(x,y)}{M} - \frac{1}{2}\iint_{\theta=\theta_2}\sigma\frac{\partial q^2}{\partial\theta}\frac{d(x,y)}{M} + \overline{\left(\frac{\partial q}{\partial\theta}\right)^2}\right) - k_h\overline{|\nabla q|^2} - \frac{\overline{q^2}}{\tau_{chem}}
\end{aligned}
\qquad (13)
$$

In this equation $M$ is the airmass of the volume between the two isentropic surfaces $\theta_1$ and $\theta_2$ and

here the overline denotes the mass-weighted average of that volume. $k_h$ and $k_\theta$ are the isentropic and diabatic diffusivities, respectively, and $\tau_{chem}$ is the chemical lifetime of the trace gas. These parameters have the meaning of mass-weighted averages for the enclosed volume.

Given an observed trace gas field, the spatially bounded tracer variance equation can be reduced to the form $f(t) = -k_\theta a(t) - k_h b(t) - 1/\tau_{chem}$ by dividing both sides of Eq. 13 by $\overline{q^2}$. Then, $f(t)$

refers to all terms on the left hand side and $a(t)$ and $b(t)$ represent coefficients of isentropic and diabatic diffusivities on the right hand side of the $\overline{q^2}$-normalized equation. The diffusivities $k_\theta$ and $k_h$ can be estimated by linear regression of the forms $f(t) = -k_\theta a(t) + c_1$ and $f(t) = -k_h b(t) + c_2$ with constant offsets $c_1$ and $c_2$.

### 3 Application

In this section the analysis framework derived in Sect. 2 is applied to estimate stratospheric stirring and mixing from the distribution of a long-lived trace gas in a numerically generated methane scenario. In the following we describe the details of the numerical experiment.





### 3.1 Experimental and numerical setup

We analyze the simulated methane with the standard version Chemical Lagrangian Model of the
Stratosphere (CLaMS v1.0). CLaMS is a Lagrangian model for modelling chemistry and transport
for trace gas species via stratospheric chemistry module (McKenna et al., 2002a), Lagrangian
advection module based on 3-D forward trajectory calculation and parametrized mixing module
driven by strong flow deformations (McKenna et al., 2002b; Konopka et al., 2004). The forward
trajectory calculation is driven by the wind data from ERA5 reanalysis (Hersbach et al., 2020) and
the total diabatic heating rates derived from ERA5 temperature tendency properties (Ploeger et al.,
2021). The configuration as well as the model initialization follows the model setup described by
Pommrich et al. (2014) with 100 km horizontal resolution and 400 m vertical resolution around the
tropopause.

The following analysis is restricted to the stratospheric region between 500 K and 1800 K. For the
numerical evaluation the modeled daily mean concentration fields are sampled on isentropic
surfaces in that region. The estimation of isentropic mixing of the trace gas needs the evaluation of
the Laplacian of its mixing ratio within isentropic surfaces. The Laplacian is calculated analytically
by expressing the two-dimensional mixing ratio field as a series of spherical harmonics truncated by
the triangular T75 truncation (e.g. Daley and Bourassa, 1978). The horizontal resolution of the
triangularly truncated approximation is coarser than the model resolution because the effective
resolution in simulated mixing ratio fields is typically reduced by diffusion.

We evaluate the stirring from the simulated methane fields using the derived diagnostic formula
(Eq. 10a and 12). In that formula, time derivatives are expressed as finite differences of two
adjacent daily mean $CH_4$ contours and air density. The other terms require the construction of the
function $m(q) = M(1)$ along isentropic surfaces from daily mean $CH_4$ fields. At each isentropic
surface the range of methane mixing ratios is linearly discretized by 31 values $q_j$ , $j = 1, ..., 31$. The
function values $m(q_j)$ are then calculated by summation of the mass located in model grid cells with
mixing ratios smaller than $q_j$. The functions $M(\dot{q})$ and $M(\dot{\theta})$ are constructed in a similar
way. For the analysis we neglect chemical sinks in the stratosphere and use the approximation
$\dot{q} = k_h \nabla^2 q$ . Diabatic heating rates and air densities are provided as output by the CLaMS model
as a regridded and processed version of the ERA5 reanalysis fields.



## 4 Results

### 4.1 Estimated diffusivities

The isentropic and diabatic diffusivities $k_h$ and $k_\theta$ for the model setup are estimated using the linear
regression approach described in Sect. 2.4. Figure 2 present the fitting effects by the forms $f(t) =$
$-k_\theta a(t) + c_1$ and $f(t) = -k_h b(t) + c_2$. As expected, the distributions of the points $(x, y)$ with $f(t)$ as $y$-
coordinate and $a(t)$ or $b(t)$ as $x$-coordinate, respectively, show linear relations. This confirms the
feasibility of the used method to estimate the diffusivities. The obtained diffusivities for the
stratospheric region bounded by 500 K and 1800 K are $k_h=0.3205\times10^5$ m$^2$/s and $k_\theta=0.0003$ K$^2$/s.

### 4.2 Reduction of diabatic effects


Contributions by diabatic motion in local temporal variations of $q$ and $m$ are shown in Figure 3. The

terms under comparison are $\dot\theta\frac{\partial q}{\partial\theta}|_{x,y,t}$ and $(\dot\theta-\hat{\dot\theta})\frac{\partial q}{\partial\theta}|_{m,t}$ , where the local temporal variation

of $m$ has been scaled by $\frac{\partial q}{\partial m}|_{\theta,t}$ . As the influence of diabatic transport increases with altitude the

reduction of such influence is more significant at high levels. For example, diabatic transport of $q$
can be as high as 400 ppb/day in the local temporal variation of $q$ but only around 100 ppb/day in
that of $m$ at 1638 K on Jan 27 of 2010. This reduction in diabatic contribution is helpful when
deriving horizontal stirring information from satellite-observed trace gases. Due to scarce
measurements vertical wind speeds in the stratosphere are highly uncertain and the wind speeds of
the modeled residual circulation vary largely between models. The tracer concentration field is
determined by 3-dimensional motion of air. The local temporal variation of $m$ is less sensitive to
vertical motion than that of $q$. Therefore, the diagnostic formula for isentropic stirring efficiently
extracts horizontal stirring from temporal variations of trace gas concentration created by 3-
dimensional motion.

### 4.3 General characteristics of stirring and mixing

Temporal variations of the trace gas mixing ratio due to mixing are proportional to the isentropic
diffusivity and the Laplacian of the mixing ratio on isentropic surfaces. Mixing reduces the spatial
inhomogeneity in the mixing ratio. The Laplacian presents maxima/minima at minima/maxima of
the trace gas mixing ratio. Mixing ratios of CH$_4$ are higher in the tropics than the polar regions as a
result of the residual circulation. At the global scale, the rising branch of the residual circulation



produces maxima by uplifting $CH_4$-rich air. Sinking branches produce minima. At smaller scales, lots of wave disturbances result in local maxima/minima of the mixing ratio in the zonal direction.

Stirring is caused by nonuniform advection along the contour of the trace gas. Stirring effects on the trace gas mixing ratio are closely related to wave disturbances, e.g. planetary wave. Figure 4 shows the horizontal distribution of the trace gas $CH_4$, and temporal variations of the mixing ratio due to

stirring and mixing at 610 K and 1164 K on Jan 27 and Jul 16, 2010. According to the distribution of the $CH_4$ mixing ratio, a wave breaking event can be recognized in a region of 100°-200° in the southern hemisphere on Jul 16, 2010. Correspondingly, temporal variations of the trace gas mixing ratio due to stirring present significant values along this wave disturbance. Isentropic stirring processes always produce filamentary structures in tracer concentration fields along the isentropic

surfaces. As mixing depends on the spatial inhomogeneity of the tracer concentration, regions where stirring is strong also show strong mixing. In contrast, there are no significant stirring effects in the northern hemisphere in July. The summer hemisphere is usually free from wave disturbances due to the presence of easterlies that prevent upward propagation of planetary waves. Similarly, strong wave disturbances occur in the northern hemisphere on Jan 27. In that instance, the polar

vortex is pushed off the pole and the northern extratropics show strong stirring.

Figures 5 and 6 show the evolution of the temporal variation of the trace gas mixing ratio at 781 K from 2009 to 2011 due to stirring and mixing. As explained above, stirring is closely related to wave disturbances. Wave disturbances in the stratosphere have seasonal cycles with stronger disturbances in the hemispheric winter and weaker ones during hemispheric summer. Temporal variations of $CH_4$

mixing ratios due to stirring present significant values in the southern extratropics during Apr-Nov as indicated by Fig. 5. This observation is consistent with numerous wave disturbances in this region during summer and the breaking of polar vortices. Temporal variations of $CH_4$ mixing ratios due to mixing in Fig. 6 show meridional maxima around the polar barrier and the subtropical barrier, and closely follow the barrier in the southern hemisphere. In the northern hemisphere only

the subtropical barrier can be recognized in 25-40° according to meridional maxima of the mixing effect.

Zonal mean distributions of $CH_4$ mixing ratio in the stratosphere are shown in Fig. 7. The zonal mean distribution is the result of the residual circulation and isentropic stirring and mixing. Air following the residual circulation rises in the tropics and sinks in the extratropics, with stronger

sinking in the winter hemisphere, and is driven by zonal forces produced by planetary wave breaking events. These wave breaking events occur in the winter stratosphere because planetary waves propagate upwards only in the westerlies and the zonal winds of the stratosphere show





dominant westerlies in the winter and easterlies in the summer (Andrews et al., 1987). Correspondingly, the rising branch of the residual circulation in the tropics biases towards the summer hemisphere.

$CH_4$ has sources at the surface and undergoes strong oxidation and photolysis reactions in the upper stratosphere and the mesosphere. As a result, upward motion increases and downward motion decreases $CH_4$ mixing ratios in the stratosphere. The rising branch of the residual circulation corresponds to the high-value $CH_4$ mixing ratio region that stretches upward in the tropics (see Fig. 7). The sinking branches indicated by the low-value $CH_4$ mixing ratios stretch downwards from the extratropics towards the polar regions. The sinking branch in hemispheric winter extends to significantly lower altitudes than the sinking branch in hemispheric summer.

The effect of the stirring term on the temporal variation of the methane mixing ratio is calculated using Eqs. 10a and 12. Figure 8 shows the zonal mean distributions of temporal variations of $CH_4$ mixing ratios due to stirring. The dominant effect of stirring is the strengthening of $CH_4$ mixing ratios in sinking branches of the residual circulation. Effects by isentropic stirring in the tropical rising branch are minor. This reflects the fact that the tropical stratosphere is weakly disturbed by extratropical waves due to the subtropical barrier (Chen at al., 1994; Neu et al., 2003). In the extratropics, the most important dynamical processes are the build-up of the polar vortex during winter and its break down during spring. There are lots of wave disturbances during these periods. The northern polar vortex starts to build up in the middle of SON (Sep., Oct. and Nov.) and breaks down around the end of DJF (Dec., Jan., and Feb.). As a result, stirring effects are large during SON and DJF in the northern extratropics. Similarly, stirring effects are large during JJA (Jun., Jul. and Aug.) and SON in the southern extratropics.

Mixing effects as shown in Fig. 9 are large in regions where strong spatial variations in the trace gas mixing ratios occur, e.g. around transport barriers. In the southern extratropics mixing effects are largest during JJA when the southern polar vortex is stable, i.e. during the period large contrast is present in the trace gas mixing ratios between the inside and the outside of the polar vortex. Similarly, mixing effects are large during SON in the northern extratropics.

### 4.4 Diagnostic of stirring and mixing of trace gas in the polar vortex

In this part the diagnostic method developed in Sect. 2 is applied to analyze effects of stirring and mixing on trace gas concentrations in the northern polar vortex during the period Nov 10, 2009 to Feb 05, 2010. This polar vortex is continuously disturbed by breaking planetary waves from lower latitudes and splits on Feb 04, 2010.





The evolution of the vortex area, the mean $CH_4$ mixing ratios in the polar vortex, and the effects of
      stirring and mixing on the mean $CH_4$ mixing ratios at 781 K are displayed in Fig. 10. The vortex
      boundary is defined as potential vorticity lines that correspond to the largest potential vorticity
      gradient with respect to the equivalent latitude. According to the results stirring mostly increases the
      mean $CH_4$ concentration in the vortex. Mixing increases the mean $CH_4$ concentrations as expected.

However, the combined effects of stirring, mixing and vortex variation mostly decreases the mean
      $CH_4$ concentration in the vortex. This means that wave disturbances tend to penetrate into the vortex
      interior but the vortex tends to avoid such penetration through movement and deformation. Also the
      vortex looses airmasses with high $CH_4$ concentration by adjusting its boundary. In a few cases the
      penetration into the vortex succeeds and increases the mean $CH_4$ concentration in the vortex.

To explain this point further, Figure 11 shows frequency distributions of $CH_4$ mixing ratios in the
      vortex, variations of the frequency distribution and relevant contributors (stirring, mixing and
      vortex variation), spatial distributions of the mixing ratios at 781 K on Dec 08 and 09, 2009, and
      Jan 28 and 29, 2010. It should be noted that predicted variations of the frequency distribution
      according to the contributors do not match the variation of the frequency distribution in the model

simulation. The reason is that only stirring, mixing and vortex variability are taken into account
      while mean transport along the tracer isolines and vertical advection are not considered. On those
      two days stirring strengthens high values (> 600 ppb) in the frequency distribution. This
      strengthening by stirring is persistent even after taking into account effects of polar vortex
      movement and deformation. This persistence indicates that wave disturbances penetrate into the

interior of the vortex. The penetration can be clearly recognized from patterns of the $CH_4$
      concentration fields on Jan 29, 2010 (see Fig. 11). On that day a long tail with high-value $CH_4$
      mixing ratios penetrates into the vortex. The pattern confirms the diagnosed effect of stirring to
      increase the methane concentration in the vortex.

## 5 Discussion

According to Nakamura (1996) the temporal variation of the trace gas mixing ratio due to stirring
      and mixing in the two-dimensional case, i.e. within isentropic surfaces, is expressed as

$$\frac{\partial q(A,t)}{\partial t} = \frac{\partial}{\partial A}\left(k_h L_e^2(A,t)\frac{\partial q(A,t)}{\partial A}\right) \ , \quad (14)$$

where $k_h$ is a constant diffusion coefficient and $L_e$ is a non-constant diffusion coefficient called
equivalent length. $A$ is the area enclosed by the contour line $q$. The variable diffusion coefficient can

amplify or block the diffusive flux along the areal variable $A$. The equivalent length is lower





bounded by the actual length of the contour (Haynes and Shuckburgh, 2000). $L_e$ is proportional to the length of the contour and the magnitude of the gradient of the trace gas mixing ratio across the contour. The equivalent length is increased by stirring and decreased by diffusion. In this expression, diffusion is included in both the constant and the variable coefficients but stirring is

included in the latter only. According to studies by Shuckburgh and Haynes (2003) and Marshall (2006), dependence of $L_e$ on $k_h$ varies according to the Pe number. The Pe number describes the relative magnitude of the advective to the diffusive time scale. $L_e^2$ scales like $k_h^{-1}$ when the Pe number is large and approaches a lower limit as the Pe number decreases. As Eq. 14 is expressed with respect to the areal coordinate, the diagnoses of the equivalent length only represents a

weighted average along the tracer contour. When applied to methane fields with maxima in the tropics, diagnoses based on the MLM theory is only useful if applied to each hemisphere separately. Alternatively, a tracer whose concentration decreases/increases monotonically from North to South can be used, e.g. the potential vorticity (Manney and Lawrence, 2016) or an artificial tracer (Allen and Nakamura, 2001).

In the diagnostic quantities developed here, Eqs. 10-12, stirring and mixing are represented by separate terms. The expression for stirring, Eq. 10b, does not depend on the diffusion coefficient $k_h$. In addition, diagnosed stirring and mixing have resolvability along the contour of the trace gas. This resolvability can be explained as following.

The expression for the stirring effect on $m(x, y, \theta, t)$ (Eq. 10b) includes two terms,

$-(\vec{v} - \vec{v}_d) \cdot \nabla m$ and $D - \hat{D}$. The first term represents the modification of $m$ by wave disturbances. Clearly the wind $\vec{v} - \vec{v}_d$ includes local information only. The gradient of $m$, $\nabla m$, depends on the local structure of the tracer contour and the magnitude of $m$ that is determined by the nonlocal distribution of the tracer isoline along the whole isentropic surface. After applying the transformation factor $\frac{\partial q}{\partial m}\big|_{\theta, t}$, as in Eq. 12, the gradient term becomes $\nabla q$ and the nonlocal

information is removed. Consequently, the first term of the expression of stirring, $-(\vec{v} - \vec{v}_d) \cdot \nabla q$, is a local term. The second term, $D - \hat{D}$, contains the nonuniform part of the divergent motion along the contour and is therefore nonlocal. Its calculation at a particular position is affected by nonlocal processes. In summary, the expression for stirring in Eq. 12 has resolvability along the tracer contour and is local to an extent that the divergent motion induced by vertical mass

convergence can be neglected.





In Sect. 4 one constant diffusivity is applied to the whole stratospheric region between the isentropic surfaces at 500 K and 1800 K. However, the isentropic diffusivity varies vertically due to vertical variations of wind velocities (Allen and Nakamura, 2001). In their scenario, the estimation of diffusivity is less challenging due to the absence of flow in and out of the region. In addition,

diffusivity has inhomgeneities in the horizontal as well as anisotropy (Konopka et al., 2005). Their results reveal that mixing occurs only in regions where flow deforms and shears strongly and diffusivity along the wind is greater than that across the wind. A more precise estimation of diffusivity could be achieved by taking flow deformation into account.

## 6 Conclusion

The modified Lagrangian-mean method (MLM) that describes the tracer evolution in the tracer contour based coordinate is transformed into a form expressed with respect to the geographic coordinates latitude and longitude. In the transformed expression temporal variations of the tracer mixing ratio by isentropic stirring and mixing are included in two separate terms. Effects by stirring and mixing can be estimated by the tracer contour and diabatic heating rates, but do not require

horizontal wind information. Diabatic heating rates are necessary because the tracer distribution is a result of 3-dimensional motion of air in the stratosphere. The developed method reduces the influence of diabatic motion on the diagnosed stirring, which is advantageous for stratospheric analysis where observations of diabatic heating are sparse. The diagnostic for isentropic stirring has resolvability along the tracer contour and only reflects the effect by local air motion to an extent

that horizontal divergent motion induced by diabatic convergence can be neglected. The method can be used to diagnose stratospheric transport based on the contour of trace gases observed by satellites. In this case limitations could come from satellite observations that are asynoptic and generally lack coverage in the polar regions.

The expression for stirring and mixing is validated in a numerical simulation by the CLaMS model

for stratospheric methane. Stirring effects diagnosed via the methane distribution are consistent with stratospheric dynamics, e.g. the isolation of the tropical stratosphere from the rest, seasonal cycles of planetary wave activities, the polar vortex and the transport barriers. Such consistency between the diagnosis and the dynamics indicates validity of the method.

The developed method is applied to diagnose stirring and mixing effects on methane concentration

in the northern polar vortex for the period from Nov 10, 2009 to Feb 05, 2010. In this period, stirring tends to increase the mean methane concentration in the polar vortex through strengthening high values in the frequency distribution of the mixing ratios. Such increasing effect is reversed by





vortex movement and deformation in most cases. Only in a few cases planetary wave disturbances can penetrate into the vortex interior and diagnosed stirring increases the mean methane concentration in the vortex.

## Acknowledgements

This research is funded by the Project 41805030 supported by NSFC.

## Data availability

ERA5 and ERA-Interim reanalysis data are available from the ECMWF. The CLaMS model data used for this paper may be requested from M. Tao (mengchutao@mail.iap.ac.cn).

## Author contribution

ZW, NH, and WT designed the study, analyzed data, and wrote the paper. MT conducted CLaMS model simulation. All coauthors commented on the paper.

## Competing interests

The authors declare that they have no conflict of interest.

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

Figures

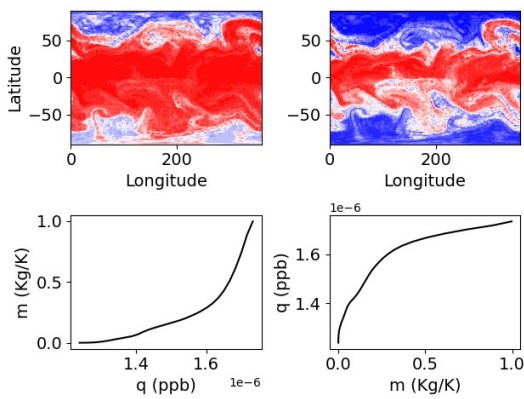

Figure 1. An example shows $q(x,y)$ (upper left), $m(q)$ (lower left), $m(x,y)$ (upper right) and $q(m)$ (lower right).



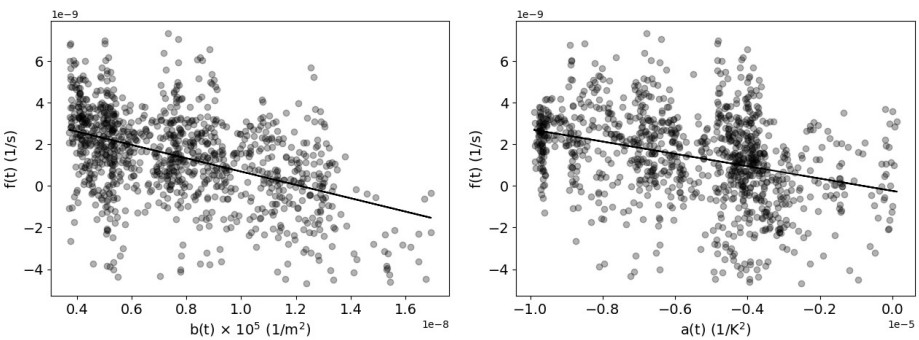


Figure 2. Linear regressions to estimate the isentropic (left) and diabatic (right) diffusivities.

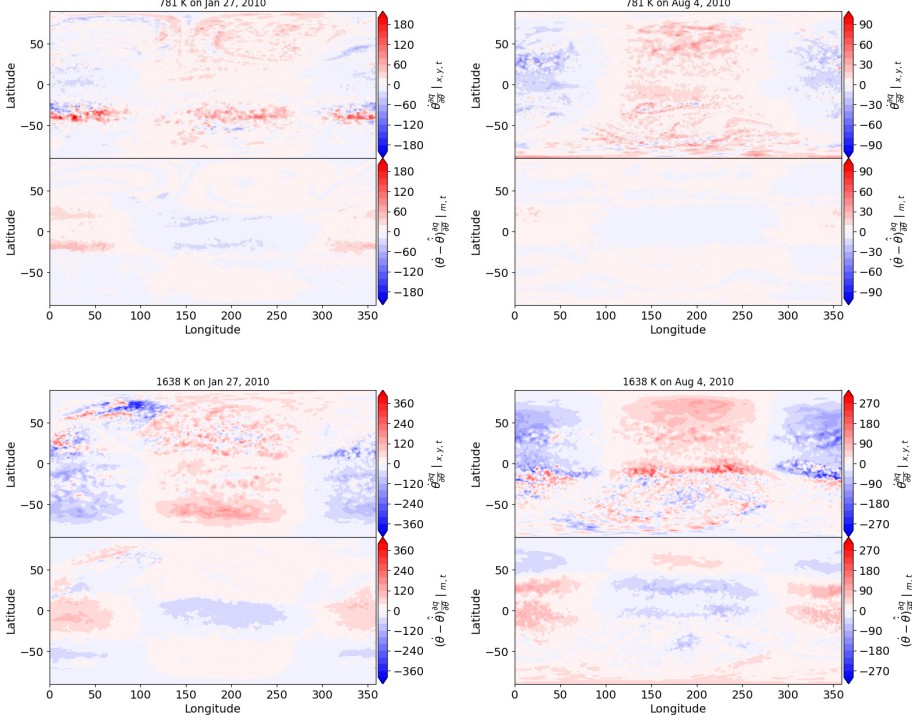

Figure 3. Examples for the reduction of the diabatic contribution to the local variation of $m$

compared to that of $q$ on different isentropic surfaces. The diabatic contribution to the local

variation of $m$ are multiplied by $\left. \dfrac{\partial q}{\partial m}\right|_{\theta,t}$ to make it comparable with diabatic transport of $q$. All

quantities are in ppb/day.

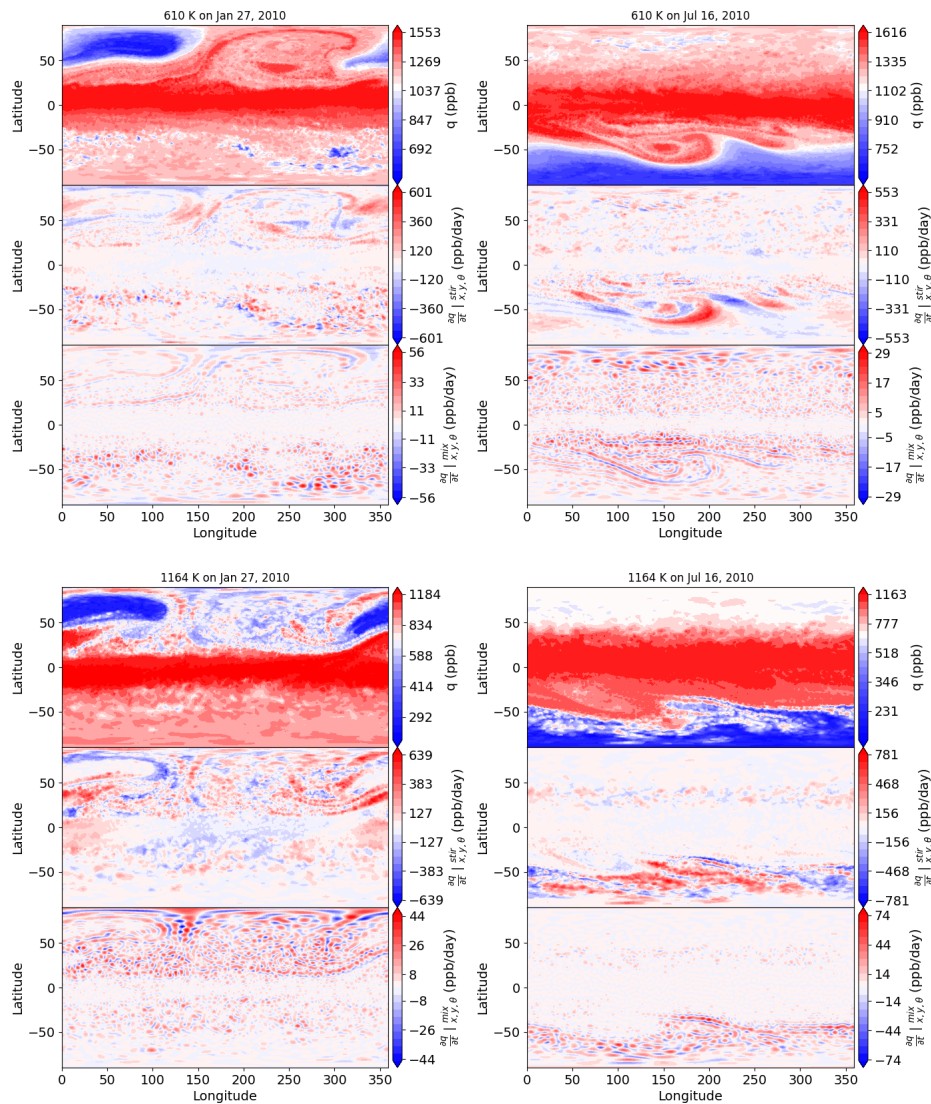


Figure 4. Snapshots of the trace gas distribution $q$ and diagnosed stirring and mixing. Stirring is significant only in the hemispheric winter while the summer hemisphere is relatively calm. It is worth noting that the distributions of $q$ and the derived stirring and mixing in particular show oscillations. These oscillations are caused by the high spatial inhomogeneity in tracer mixing ratios, which naturally occur in Lagrangian transport models that have small diffusion.




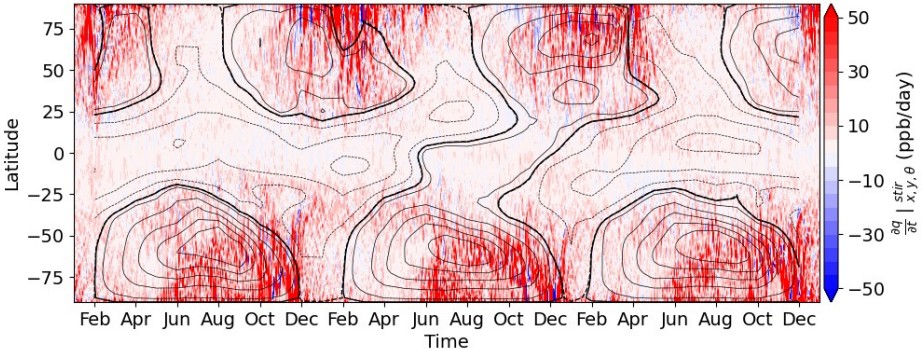


Figure 5. Evolution of zonal mean temporal variations of the trace gas mixing ratio due to stirring, and zonal mean zonal wind (black lines with solid ones for westerlies, dashed ones for easterlies and bold line for zero wind, increment is 15 m/s) during 2009 to 2011.

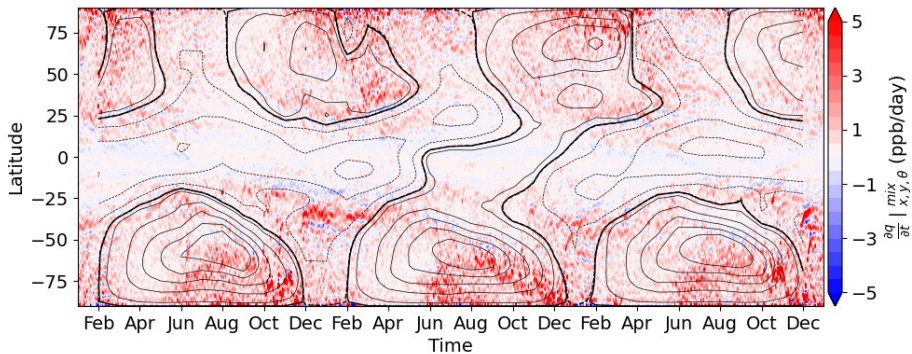


Figure 6. Same as Fig. 4 except for evolution of zonal mean temporal variations of the trace gas mixing ratio due to mixing.







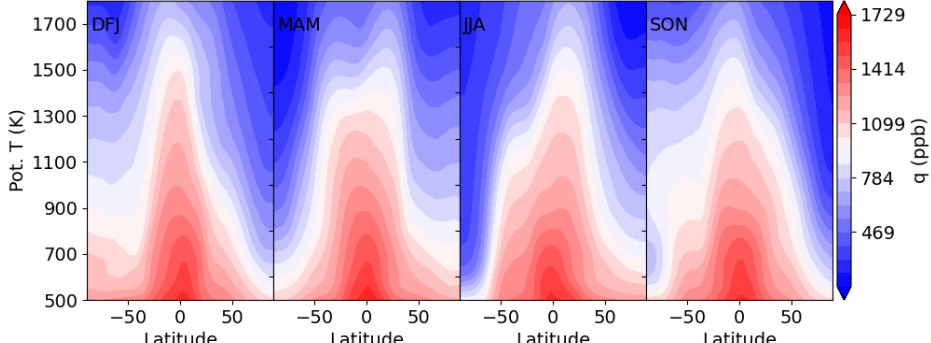

Figure 7. Zonal mean distributions of mixing ratios of the long-lived trace gas CH$_4$ modeled by CLaMS. Each column represents a seasonal average for the period 2009 to 2011.


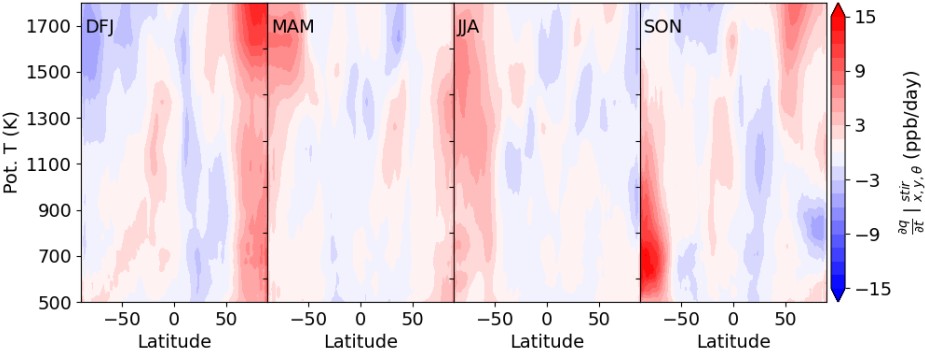

Figure 8. Zonal mean distributions of temporal variations of the trace gas mixing ratio due to stirring. Each column represents a seasonal average for the period 2009 to 2011,.


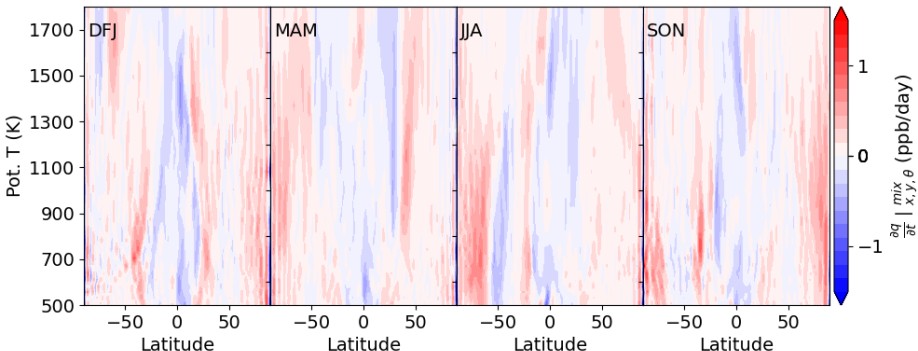

Figure 9. Same as Fig. 7 except for zonal mean distributions of temporal variations of the trace gas mixing ratio due to mixing.

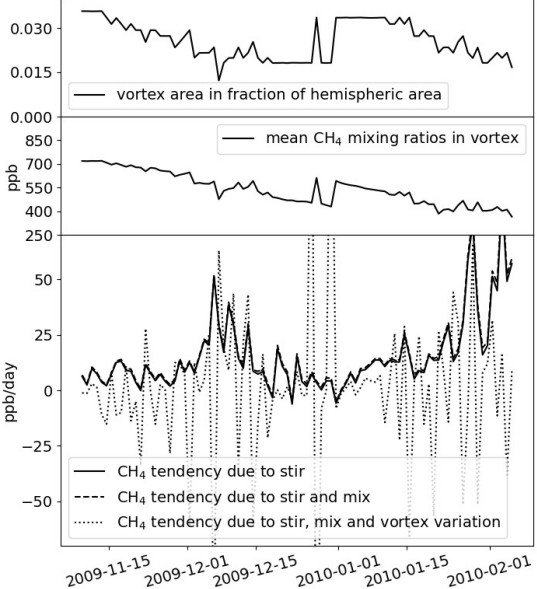


Figure 10. Evolution of the northern vortex area (top), mean $CH_4$ mixing ratios in the vortex (middle) and temporal tendencies of the mean $CH_4$ mixing ratios due to stirring, mixing and vortex variation (bottom) at 781 K from Nov 10 2009 to Feb 5, 2010. The vortex boundary is defined as the contour that corresponds to the largest potential vorticity gradient with respect to equivalent

latitudes.



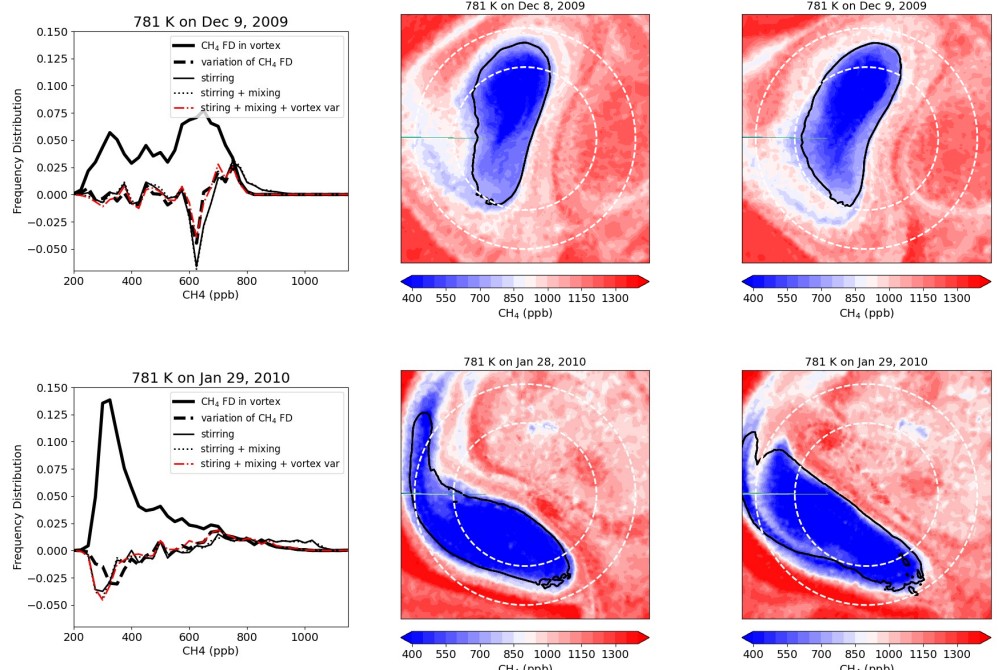


Figure 11. CH$_4$ frequency distribution in the northern polar vortex, its variation and contributors along the 781 K surface (left). Middle and right: CH$_4$ mixing ratio contours, latitudinal circles of 60°N and 45°N (white dashed lines) and vortex boundary (black solid lines, defined as the contour

that corresponds to the largest potential vorticity gradient with respect to equivalent latitudes) at 781 K on Dec 8-9, 2009 and Jan 28-29, 2010.