# Peer review of "Estimation of isentropic stirring and mixing and their diagnosis for the stratospheric polar"

_Atmospheric Chemistry and Physics, 2021_

## Referee Comment (RC1)

This work aims at diagnosing transport and mixing in the atmospheric stratified fluid by separating these notions in a 3D formulation where the variables are time, potential temperature and tracer value or the mass of fluid within a tracer contour.

Although the aim is interesting and the derivation contains several ideas that might be pursued usefully, the paper is very confusing and unconvicing and I cannot recommand it for publication.

Section 2.1 is a derivation of a stirring term from a series of manipulations of the tracer equation which are arbitrary and to a large extend circular. The final equation (11) can be written directly from (2) and (3) with a "stirring" term forged to replace the average source $\widehat{q}$ and heating $\widehat{\dot{\theta}}$ by local terms, avoiding the detour by $S_m$. A further replacement of $\partial M(\dot{\theta})/\partial\theta$ by $\widehat{D}$ is necessary but cannot be interpreted in the same way.

A number of notions are defined but each time with no clear justification and accompanying sentences that only restate the arbitrary definitions or are questionable interpretations. This is true for (8), (9) and (10). In the definition of $D$, $v_d$ is by no way the horizontal divergent wind as it is suggested and I wonder in which $\dot{\theta}\partial m/\partial\theta$ is a "reduction by the horizontal expansion associated with vertical diabatic motion". In the same way, there is nothing showing why (10) should be a definition of local isentropic stirring. Justifications accompanied by relevant well built examples should be provided.

Equation (12) is somewhat antagonic to the work of Nakamura and followers who showed that stirring is an irreversible process that leads to mixing whatever large or small is the actual diffusive process that performs the final regularization. The irreversibility here is of the same nature as that of the kinetic theory of gases which is also based on a reversible set of equations. It was even shown that the effective diffusivity resulting from mixing does not depend on the small-scale diffusion (Shuckburgh Haynes, 2003). Therefore, mixing cannot be separated from stirring and this issue fragilizes the rest of the work.

The confusion is increased by defining a diffusion which is only isentropic while it is later realized that a vertical diffusion is also needed.

Section 2.2 adds to the confusion by pretending to derive an estimate of the diffusion from Nakamura 1966 revisited by Leibensperger and Plumb (2014) while in fact Wang et al. JGR, 2020) - by the way missing in the reference list - is only providing a spatial average estimate based on the gradient dissipation. As ClAMS does not contain any explicit diffusion but performs mixing by merging or splitting the parcels, this can be a crude way to estimate an equivalent numerical diffusivity.

In section 2.2, the notation $M$ is used with a very different meaning than in section 2.1. This suggests this work has hastely assembled different and unrelated parts.

The fit to the experimental data in 4.1 is very poor and the slopes that determine the diffusivity are very badly determined. In addition, there is no justification of performing a separate fit for the two diffusivities. One can even consider that the horizontal diffusivity is a product of the vertical diffusivity

and the squarred ratio between vertical shear and horizontal strain as explained in Haynes & Anglade (JAS, 54, 1121-1136, 1997).

Figure 3 based on a single day is hardly a proof of concept and it is even not clear that the reduction occurs everywhere in the domain due to the very bad choice of color scales.

Figure 4 shows two examples in winter and summer. The comments are well known generalities about methane, transport and mixing in the stratosphere but nothing is said on what we learn here from the mixing and stirring fields. It is actually very difficulst to make sense of this figure and not only because the color scale is again very poorly chosen (probably the default choice of the plotting program).

Then, and very surprisingly, figs 5 to 9 show zonal average results. It is a bit difficult to understand why a work that is focused on defined a longitude dependent diagnostic of mixing and stirring produces results in a zonal mean for which all this work is basically useless. It is not even clear that equivalent latitude averaging is performed and no mention is made of the previous work of Shuckburgh et al. (JGR, 2001, doi:10.1029/2000JD900664) to which this should be obviously compared. It seems here that the signal is limited to the effect of the polar vortex and no QBO related variability is obtained in the tropics. This is somewhat surprising owing to the abundant literature on transport and mixing modulation by the QBO in the tropics.

Figure 10 shows some results averaged over the polar vortex, therefore again without any longitude dependency. The dash line mentioned in the legend seems missing on the plot. It is hard to understand why the added effect of stirring and mixing is in the opposite way of the two components. The accompanying comment suggests that taking into account the deformation by doing contour averages has not been done here and this again against the motivation of this work. Figure 11 does not clarify this issue.